# Learning Curve for Short-Stem Total HIP Arthroplasty through an Anterolateral Approach

**DOI:** 10.3390/medicina59050832

**Published:** 2023-04-24

**Authors:** Martin Bischofreiter, Christina Kölblinger, Thomas Stumpner, Michael Stephan Gruber, Michael Gattringer, Harald Kindermann, Georg Mattiassich, Reinhold Ortmaier

**Affiliations:** 1Department of Orthopedic Surgery, Ordensklinikum Barmherzige Schwestern Linz, Vinzenzgruppe Center of Orthopedic Excellence, Teaching Hospital of the Paracelsus Medical University Salzburg, 5020 Salzburg, Austria; koelblinger.christina@gmail.com (C.K.); michael.gattringer@gmx.net (M.G.); reinhold.ortmaier@ordensklinikum.at (R.O.); 2Department of Orthopedic and Trauma Surgery, Clinic Diakonissen Schladming, 8970 Schladming, Austria; georg.mattiassich@diakonissen.at; 3Department of Marketing and Electronic Business, University of Applied Sciences Upper Austria, 4400 Steyr, Austria; harald.kindermann@fh-steyr.at

**Keywords:** hip arthroplasty, short-stem, learning curve, resident, anterolateral approach

## Abstract

*Background and Objectives*: Short-stem total hip arthroplasty has become increasingly popular in recent years. While many studies have shown excellent clinical and radiological results, very little is known about the learning curve for short-stem total hip arthroplasty through an anterolateral approach. Therefore, the aim of this study was to determine the learning curve for short-stem total hip arthroplasty among five residents in training. *Materials and Methods*: We performed retrospective data analysis of the first 30 cases of five randomly selected residents (*n* = 150 cases) with no experience before the index surgery. All patients were comparable, and several surgical parameters and radiological outcomes were analyzed. *Results*: The only surgical parameter with a significant improvement was the surgical time (*p* = 0.025). The changes in other surgical parameters and radiological outcomes showed no significant changes; only trends can be derived. As a result, the correlation between surgical time, blood loss, length of stay, and incision/suture time can also be seen. Only two of the five residents showed significant improvements in all examined surgical parameters. *Conclusions*: There are individual differences among the first 30 cases of the five residents. Some improved their surgical skills faster than others. It could be assumed that they assimilated their surgical skills after more surgeries. A further study with more than 30 cases of the five surgeons could provide more information on that assumption.

## 1. Introduction

Total hip arthroplasty (THA) is one of the most successful orthopedic surgeries [1]. The main indication for THA is osteoarthritis of the hip, the most common joint disease in elderly patients [2]. With the expansion of indications, the number of THAs, in general, is increasing, and it has also increased among younger patients [3]. The latter is expected to have a higher risk of future revision due to their lifestyle and sports activities. Studies have shown a revision rate of 6% among patients under 55 years old. Some of those require a change in the femoral stem. As a result, there is a need for a prosthesis that preserves the femoral bone [2,4,5].

Cementless standard stems are the golden standard for THA. However, short stems have become more and more popular globally in recent years. The main reason for this development is that short stems are bone-sparing and can be implanted with less soft tissue trauma [6,7].

Compared to long-stem hips, the short ones are implanted in the metaphysis. Due to the shorter fixation area, an accurate press fit is necessary to avoid secondary migration of the stem. A further advantage is the possibility of a wide range of varus and valgus alignments during implantation. It is postulated that femoral neck-sparing prostheses provide better anatomic reconstruction [6,8].

All these factors lead to different techniques compared to conventional straight cementless stems [8,9]. Each surgical procedure undergoes a learning curve. The accurate press-fit implantation and a wide range of varus and valgus positionings could cause a different and maybe more distinct learning curve, especially for young and inexperienced surgeons. Little is known about the short-stem learning curve through an anterolateral approach. Therefore, we analyzed the first thirty short-stem THAs of five resident surgeons.

## 2. Materials and Methods

This study is a retrospective comparative study where 150 patients who fulfilled the inclusion criteria were evaluated. The patients were identified through a prospectively gathered database of all patients that underwent THA between 2015 and 2021 through an anterolateral approach. In all patients, a cementless, curved short-stem and a monobloc press-fit cup (Optimys/Vitamys, Mathys AG, 2544, Bettlach, Switzerland) were used for primary THA.

The average overall age was 66.8 (SD 11.45) years. The average overall BMI was 28.23 (SD 4.61). Out of all patients, 84 (56%) were female, 66 (44%) were male, and 78 (52%) were operated on the right side, whereas 72 (48%) had surgery on the left side. The mean ASA score was 2. Overall, 129 of the patients (86%) suffered from pre-existing illnesses, with hypertension being the most common. The main indication for the primary THA was Coxarthrosis (*n* = 143, 95.3%). Seven (4.7%) of the patients were operated on due to femoral head necrosis, while three (2%) had a dysplastic acetabulum. Details for patient demographics are shown in Table 1.

Inclusion criteria are primary short-stem THA through an anterolateral approach performed by one of the five residents. Patients with post-operative complications not deriving from a THA were excluded, as well as those with pre-operative neurological problems or those who had had hip surgery before.

In general, the length of stay was defined as the time patients spent at the orthopedic department after surgery or at other departments due to treatments of complications resulting from the total hip arthroplasty.

Five residents who had started their training were randomly chosen by picking their names out of a box. Their first 30 primary short-stem THAs were then examined. None of the residents had any surgical experience in endoprosthetic surgery, and all of them were in the first year of their orthopedic surgery residency. The mean time until the beginning of their residency and the index surgery was 6.4 months (SD 3.64). The patient demographics (age, sex, BMI, laterality, indication for operation, pre-existing illnesses, ASA score, CCD angle), as well as operation data (blood loss, transfusion rate, surgical and anesthesia time, length of hospital stay, 90 day admission rate and complications) and the radiological outcomes (cup inclination, pre-operative planning and post-operative outcome of the cup and stem sizing), were collected directly from the hospital’s data information system. For each resident, subgroups were analyzed by dividing the 30 THAs into three groups (1–10, 11–20, and 21–30). All five residents had never performed a short-stem THA before the index surgeries. Hence, the learning curve was evaluated from the very beginning of clinical care, making all residents equally comparable.

### 2.1. Surgical Technique

The anterolateral approach was performed in the supine position. An oblique incision was made in the direction of the gluteal muscle. The beginning of incision started at around 1–2 cm distal of the greater trochanter and ended at approximately 2 cm lateral and distal to the anterior superior iliac spine. In some cases, however, the incision had to extend more proximal and distal. The fascia was located and opened dorsal of the M. tensor fasciae latae. The intermuscular interval between the gluteus medius and tensor faciae latae was openend. After identifying the joint, a ventral capsule resection was performed. In the next step, the femoral neck was resected according to the pre-operative planning, and the head was extracted. After sufficient exposure to the acetabulum and under permanent intraoperative radiological control, the acetabulum was reamed down to the planned depth, and the cup was implanted. Then the operation table was extended to about 15° in order to provide a hyperextension position. Afterward, the leg was rotated externally, and adduction was performed. The medullary canal was opened, and again under radiological control, the femoral bone was prepared with rasps according to the planned size. Finally, after trial reduction, the determined stem and head were implanted [10,11].

### 2.2. Calculation of the Blood Volume

In order to calculate the blood volume, we used the formula of Moore, where different values (ml/kg body weight) depending on the BMI are used to calculate the blood volume (BV) [12]. The values according to gender and BMI are shown in Table 2 below [12,13].

### 2.3. Calculation of the Blood Loss

For this study, the formula of Bourke was chosen to calculate blood loss (*BL*) [14]. This formula uses the pre- and post-operative hematocrit for calculating *BL*. The formula is shown below [13,14].
$$BL=BV\times \left({Hct}_{0}-{Hct}_{t}\right)\times (3-{Hct}_{mean})$$

*Hct*_0_ = pre-operative hematocrit

*Hct_t_* = any given post-operative hematocrit

*Hct_mean_* = the average between the pre- and post-operative hematocrit

### 2.4. Classification of Peri- and Post-Operative Complications

In order to evaluate the peri- and post-operative complications, all recorded complications within 90 days after surgery were incorporated. The complications were split into surgery-related and medical complications. Surgery-related complications are fractures, periprosthetic joint infections, nerve lesions, wound dehiscence, and hematoma, which need interventions (e.g., puncture) [15].

The medical complications were classified into five grades according to Sink et al. Grade I includes all complications where no change in the post-operative routine was needed besides the administration of antiemetics, antipyretics, analgetics, diuretics, and electrolytes. Complications requiring medication other than in grade I are categorized as grade II. Grade III represents complications that need invasive surgical treatment but do not come with any long-term morbidity other than those linked to grade IV, which includes long-term morbidity and life-threatening complications. Grade V is defined as the patient’s death. Due to the orthopedic adaption, grade IV also includes femoral head and acetabulum osteonecrosis, permanent nerve or vascular injury, as well as pulmonary embolism [16].

### 2.5. Radiological Outcomes

To evaluate radiographic outcomes, pre- and post-operative radiographs were analyzed. Planned stem and cup size were compared to post-operative stem and cup size, as well as cup inclination. For stem and acetabular sizes, one size larger/smaller was considered a normal deviation and, therefore, “correctly planned.” From a deviation of two sizes, the planning was judged to be faulty. The acetabular inclination is considered “normal” if its angle is between 40 and 50°, “valgus” or “varus” if its angle is >50° or <40°, respectively.

## 3. Statistical Analysis

The analysis is descriptive. Where the number of cases permitted, significance tests have been performed according to the respective scale level to be able to identify possible correlations in an explorative manner.

Metric parameters are represented by means and standard deviations or median and minimum–maximum categorical parameters by absolute and relative frequencies. To determine group differences, either *t*-tests, analysis of variance, or chi-square tests are performed depending on the scale level and the number of groups. Significant results are used only to derive hypotheses that could subsequently be proven in separate studies.

Statistical analyses were performed using SPSS version 28 and R version 4.0.4. and a standard *p*-value of ≤0.05 was used.

## 4. Results

### 4.1. Surgical Parameters

The average length of stay was 6.93 days (SD = 2.43). During the first 10 cases of all five residents, the mean length of stay was 7.02 days (SD = 2.63), which decreased to 6.88 days (SD = 1.97) during the second and to 6.90 days (SD = 2.70) during the third ten cases (*p* = 0.954).

The mean surgical time was 87.8 min (SD = 18.28). The surgical time increased from a mean value of 87.3 min (cases 1–10; SD = 19.28) to 92.9 min (cases 11–20; SD = 14.15) and then decreased to 83.2 min (cases 21–30; SD = 19.91). However, there is a significant decrease in surgical time from 87.8 to 83.2 min when the values of all 150 cases are summed up, as seen in Table 3 (*p* = 0.029). This trend also corresponds with the anesthesia time (Table 3).

During the first ten cases, the mean blood loss changed from 1102 mL (1–10 cases) to 1274 mL (11–20) and 1307 mL (21–30), respectively (*p* = 0.263). For all 150 cases, the blood loss was 1227 mL on average (SD = 671.43).

With three blood transfusions among the first two groups and two transfusions among the third group, no significant change is visible with this surgical parameter. The blood transfusion rate in our study was 5.3%.

### 4.2. Complications

In total, 30 major and minor complications occurred. Eight were recorded as surgical complications and twenty-two as medical complications of grades I–III according to the Clavien–Dindo classification. The distribution of the complications was similar across all three groups. With 9 (18%) complications among the first 10 cases, 10 (20%) among the second, and 11 (22%) among the third ten cases, neither a significant increase nor decrease was shown. The surgical complications included one bleeding, one wound secretion, three seroma/hematomas which required punction, two fissures/fractures, and one patient who had a leg length discrepancy of >1 cm. For an overview of all data, see Table 3.

In the 90 days after THA, none of the 150 patients were readmitted to the hospital because of complications in relation to the surgery.

### 4.3. Radiological Outcomes

Among all 150 cases, a total of 118 (78.6%) cups were implanted with an inclination angle of 40–50°, 15 (10%) with an angle <40°, and 17 (11.3%) with an angle >50°. In general, among the first 10 cases of all surgeons, the cups were implanted more with a varish angle (*n* = 8, 16%) and less with a valgus angle (*n* = 5, 10%) if they were not in the normal range. Among the last 10 cases, it was the opposite, and more valgus (*n* = 7, 14%) than varish (*n* = 3, 6%) angles were implanted.

The correctness of pre-operative planning for the stem size was considerably constant too. In 43 (86%), 45 (90%), and 41 (82%) cases among the first, second, and third groups, respectively, the planned stem size was implanted. Moreover, a size larger/smaller is considered a normal deviation and thus correctly planned. Twelve times (8%), the stem size was planned too small and nine times (6%) too large.

Among the three groups, 44 (88%), 45 (90%), and 43 (86%) times, the planning of the cup size was correct, while 6 times (4%), it was planned too small and 12 times (8%) too large. For a better overview of the data distribution, see Table 4.

## 5. Discussion

In this study, we evaluated the learning curve of five residents in training who performed a total hip arthroplasty through an anterolateral approach with a short-stem hip. There was no significant general learning curve among all five residents. When summing up all residents, a significant positive learning curve regarding surgical time was found even though individual differences were detected.

For this trial, the residents in training were chosen randomly. They all had not performed total hip endoprosthesis before their index surgery. We decided on an examination of the first 30 cases as the current literature often suggests this number. However, there is no validation for it being the most representative one for the learning phase [17,18].

The patients can be considered representative of the general population as their average BMI of 28.23 (SD = 4.61) is the same as the average of the general Austrian population [19]. Their average age of 66.8 years (SD = 11.45) is only slightly above the mean age for THAs in the United States in 2014 [20]. With a mean ASA score of 2, most patients had some mild systemic diseases without functional limitation, such as active smoking, hypertension, controlled diabetes mellitus, or obesity (BMI > 30) [21].

The summarized surgical time of all five residents showed a significant improvement (*p* = 0.029). Loweg et al., who compared the same implant among senior consultants, consultants, and residents, showed mean surgical times of 34.7 min (SD 9.7 min, range 19–59 min), 52.9 min (SD 14.6 min, range 26–87 min) and 62.7 min (SD 11.7 min, range 43–82 min) [3].

Similar to our study, the residents in training performed the surgical procedures with a senior consultant or consultant as a first assistant within the scope of a certified hospital for high-volume endoprosthetic surgery. Even though there is a difference of over 20 min between the residents, an improvement over time can be observed in both studies [3].

A possible explanation for the 20 min difference between our study´s residents and the ones in Loweg et al.´s study is that the latter performed only 9% (*n* = 26) of the total amount of operations (*n* = 287). It is also not clear if the surgical procedures were the very first ones or if those residents already had surgical experience in THA.

Unlike the anesthesia and surgical time, the hospital stay shows a continuous decrease over all three groups if all 150 cases are summed up, but this is still not significant (*p* = 0.46). If the trends of the residents are examined separately, only residents 2 and 3 showed improvement. The length of stay for residents 1, 4, and 5 did not show positive progress. This trend could have been expected for residents 4 and 5 as their anesthesia and surgical time increased as well, but the result was rather surprising for resident one.

Among the three case groups, 9 (18%), 10 (20%), and 11 (22%) complications occurred among the 150 cases, which is not a significant change (*p* = 0.510). The number of surgical complications, in particular, shows no development (*n* = 3 (6%), *n* = 2 (4%), and *n* = 3 (6%) records among the three groups). The same conclusions can be drawn when analyzing the residents individually. The number of complications among the three case groups of every single resident remains quite constant. Only resident 2 shows a higher rate in his second group, although only medical and no surgical complications occurred.

Compared to another study, the learning curve among experienced surgeons when learning a new operating method (short stem THA) was evaluated. The focus of the study was on the complication rate. The study comprises 182 short-stem THAs and demonstrates a fracture rate of 2.9%, whereas 80% of them occurred in the first 30 cases. With a fracture rate of 1–3%, our study showed good results compared to Padilla et al [8].

According to the current literature on THAs, the average blood loss is estimated to be between 1188 and 1651 mL [13]. Some studies even report values up to 1800 mL, with a transfusion rate of 10–69% [22,23]. Higher blood loss and transfusion rates correlate with longer hospital stays and higher complication rates. Post-operative anemia is associated with hypertension and tachycardia, chest pain, fatigue, and many other complications, such as myocardial infarction. The thereby needed blood transfusions also pose great risks for patients. The transfusion-related complications include systemic infections (e.g., HIV, Hepatitis) as well as local infections due to transfusion-triggered immunomodulation; venous thromboembolism, pneumonia, sepsis, and mortality are reported frequently [13,24,25,26].

Therefore, an improved calculation of blood loss would be beneficial for estimating possible complications. Furthermore, the transfusion rate could be reduced by using cell savers or pharmacological agents [13].

Unfortunately, there is no gold standard for estimating perioperative blood loss. Different formulas are proposed by the present literature, such as the formulas of Liu, Mercuria, Bourke, Ward, Lisander, Gross, or Meunier. All those formulas are based on the patient´s blood volume and pre- and post-operative hematocrit, but they still show a wide range of different results. However, the evaluation of blood loss by using formulas is important since direct intraoperative measurements are not able to record hidden blood loss or post-operative rebleeding [13].

The blood loss of the one hundred fifty cases shows an increasing trend if all five residents are summed up. As *p* = 0.288, the course can only be seen as a tendency as it is not a significant change. Similar to the other surgical parameters, residents 1, 4, and 5 show impairment over time. The learning curve of resident 2 remains constant, and resident 3 is the only one with an improvement. Since the complication rate remains quite constant, the relationship with blood loss cannot be proven, contrary to the relationship between blood loss and length of stay, which can be proven.

With three (6%) blood transfusions among the first two groups and two (4%) transfusions among the third group, no significant change is visible with this surgical parameter. Most blood transfusions were administered among the patients of resident 4. This correlates with the complication rate. One of his patients received two transfusions because of post-operative bleeding, and the other patient received two transfusions because of post-operative hemoglobin deficiency. Apart from that, no connection between the administered transfusions and other surgical parameters was found. The transfusion rate in our study, 5.3%, is lower than reported in the current literature. The study of Hochreiter et al. reports a transfusion rate of 8%, meaning 10 out of 124 patients receiving short stems needed a blood transfusion. In the control group receiving straight stems, it was even higher, with 15.6% needing a blood transfusion (22 out of 141 patients) [27]. Some studies even report higher blood loss and transfusion rates, ranging from 10 to 69% for short stems [22,28]. Moreover, in the first 90 days after THA, none of the 150 patients were readmitted to the hospital because of complications in relation to the surgery.

All five surgeons showed similar data in terms of cup inclination and no significant improvement or worsening (*p* = 0.487). There were 37 (74%), 41 (82%), and 40 (80%) cups implanted with a correct angle of between 40 and 50°. At the beginning of the learning phase, residents tended to implant acetabular cups at a varus inclination angle, and toward the end, they tended to implant acetabular cups at a valgus angle. A possible explanation for this observation could be that the residents tried to improve the varus inclination, but due to having less experience, they overcorrected them toward the end. There was little difference between the individual physicians and their three groups. Nevertheless, resident 3 showed the most correctly implanted acetabular cups (27; 90%). Resident 5 only implanted 20 (66.6%) cups in the right position.

The planned stem size was installed in 43 (86%), 45 (92%), and 41 (41%) cases. Overall, the stem size was planned 12 (24%) times too small and 9 (18%) times too large. Again, no significant change is seen (*p* = 0.612). Surgeons 2 and 5 had the best ratios between planned and implanted socket size, with one size larger/smaller being considered a normal discrepancy and, therefore, correct.

Similar to the stem size data, the pre-operative planning of the cup size also showed no singular veining over time (*p* = 0.819). The planned cup size was actually implanted in 44 (88%), 45 (90%), and 43 (86%) cases. Again, pre-operative planning matched the implanted cup size most often in residents 2 and 4 (*n* = 28). Unlike the study by Loweg et al., no significant learning curve was found here in relation to pre-operative planning [3].

## 6. Conclusions

When observing the five residents individually, only two of them (residents 2 and 3) showed a positive learning curve in all examined surgical parameters. No improvement was visible among the other residents. When analyzing all 150 cases of the five residents together, the only parameter with a significant improvement was surgical time, which was reduced. One possible reason for the lack of improvement could be that the learning curve was not fully visible within 30 cases. What we see is that there were big individual differences among the residents, and as expected, some of them improved their surgical skills faster than others. Perhaps they assimilated their surgical skills after more surgeries. A further study with more than 30 cases of the five surgeons could provide information on this. Furthermore, an evaluation of the learning curve of these residents in another operating method could be interesting (e.g., in total knee arthroplasty). As a result, one could see whether or not the learning curve of the current examination shows a similar trend. All in all, further studies need to be conducted to confirm the results of our study, as there was a small number of cases.

## Figures and Tables

**Table 1 medicina-59-00832-t001:** Patient demographics of all five surgeons. The cases of the surgeons are divided into three groups (first 10, second 10, and third 10 cases).

	Surgeon 1			Surgeon 2			Surgeon 3			Surgeon 4			Surgeon 5		
Cases	1–10	11–20	21–30	1–10	11–20	21–30	1–10	11–20	21–30	1–10	11–20	21–30	1–10	11–20	21–30
Age (mean + SD)	67.2 (8.34)	67.8 (10.29)	67.6 (10.29)	72.4 (6.88)	69.9 (16.29)	64.5 (16.29)	72.0 (10.22)	65.2 (11.57)	65.8 (11.57)	68.7 (9.90)	60.3 (12.32)	63.8 (12.32)	66.0 (11.56)	70.7 (12.46)	60.1 (12.46)
BMI (mean + SD)	29.7 (4.35)	28.84 (4.02)	28.38 (4.02)	29.02 (6.67)	29.43 (4.82)	26.69 (4.82)	28.99 (4.69)	27.86 (4.52)	26.14 (4.52)	25.95 (4.20)	29.17 (4.18)	25.82 (4.18)	28.83 (4.86)	28.87 (3.42)	29.23 (3.42)
Gender [m|w]	2|8	4|6	4|6	4|6	6|4	5|5	5|5	3|7	4|6	4|6	8|2	3|7	3|7	6|4	4|6
Laterality [l|r]	4|6	5|5	2|8	3|7	3|7	3|7	6|4	5|5	5|5	4|6	5|5	5|5	7|3	7|3	8|2
CCD-angle (mean + SD)	130.21 (4.63)	127.02 (5.10)	129.44 (4.69)	127.43 (6.52)	128.25 (4.82)	126.21 (3.78)	128.55 (4.97)	126.43 (4.44)	130.82 (3.10)	131.19 (3.39)	130.43 (2.93)	131.21 (4.26)	134.13 (5.36)	129.08 (5.64)	130.11 (4.89)
ASA															
ASA 1	2	1	1	1	1	0	2	0	2	1	2	3	1	0	0
ASA 2	5	6	6	8	7	8	4	7	5	8	6	5	6	6	8
ASA 3	3	3	3	1	2	2	4	3	3	1	2	2	3	4	2
Diagnosis															
Coxarthrosis	9	10	9	10	10	10	10	10	9	10	10	10	9	9	9
Femoral head necrosis	2	0	1	0	0	0	0	0	1	0	0	0	1	1	1
Dysplasia	0	0	0	0	0	0	0	0	0	2	1	0	0	0	0
Osteolytic metastases	0	0	0	0	0	0	0	0	0	0	0	0	0	0	0
Pre-existing illness															
Smoking	0	2	2	1	0	1	0	2	2	1	1	2	2	2	1
Obesity	0	0	1	1	0	0	0	0	1	1	1	0	0	1	0
Diabetes mellitus	0	3	0	3	2	0	2	2	2	1	2	0	1	1	0
Neoplastic diseases	1	2	3	1	2	2	1	0	0	1	0	1	2	0	4
Hypertension	3	3	5	6	3	5	3	3	5	5	5	5	4	5	5

**Table 2 medicina-59-00832-t002:** Blood volume (ml/kg) depending on the BMI.

BMI	Male	Female
Obese	60 mL/kg bodyweight	55 mL/kg bodyweight
Normal	70 mL/kg bodyweight	65 mL/kg bodyweight
Thin	65 mL/kg bodyweight	60 mL/kg bodyweight

**Table 3 medicina-59-00832-t003:** Surgical parameters of all five surgeons split into three groups of ten cases each.

Cases	1–10	11–20	21–30	*p* Value
Blood loss + SD	−1101.9 (564.2)	−1273.9 (646.9)	−1306.8 (780.6)	0.263
Transfusion rate	3	3	2	
Surgical time + SD	87.3 (19.3)	92.9 (14.1)	83.2 (19.9)	0.029
Anesthesia time + SD	2:31:11 (1271.7)	2:38:41 (1280.6)	2:28:28 (1393.7)	0.058
Length of stay + SD	7.02 (2.63)	6.88 (1.97)	6.90 (2.70)	0.954
Complication				
No complications	41	40	39	0.5156
Surgical complication	3	2	3	
Medical complication grade I	4	7	5	
Medical complication grade II	0	0	2	
Medical complication grade III	2	1	1	

**Table 4 medicina-59-00832-t004:** Radiological outcomes of all five surgeons split into three groups of ten cases each.

Cases	1–10	11–20	21–30	*p* Value
Cup inclination	n.a. = 1	n.a. = 3	n.a. = 5	
Normal (40–50°)	37	40	35	0.255
Varus (<40°)	8	4	3	
Valgus (>50°)	4	3	7	
Stem size	n.a. = 1			
Stem size according to pre-operative planning	43	45	41	0.627
Stem size planned too small	4	3	4	
Stem size planned too big	2	2	5	
Cup size	n.a. = 1			
Cup size according to pre-operative planning	43	45	43	0.736
Cup size planned too small	1	2	3	
Cup size planned too big	5	3	4	

## Data Availability

The data presented in this study are available on request from the corresponding author.

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
