# Peer review of "Learning Curve for Short-Stem Total HIP Arthroplasty through an Anterolateral Approach"

_medicina, 2023, doi:10.3390/medicina59050832_

Round 1

Reviewer 1 Report

Good topic, but I have a few corrections.

Introduction

-The 'Revision' in lines 34-35 has several causes. It is possible to change only the liner without doing the stem side, and in some cases, only the cup is replaced. So the word 'therefore' is overkill. Make lines 35-36 more natural by adding another sentence.

Materials and Methods

-Please describe the study design and level. For example, something like " This study is a retrospective comparative study evaluating the".

-In line 58-61, please indicate male, female, right side, left side, type of disease, etc. together with the number (%).

-In lines 67-71, was the patient's overall stay presented? Given the flow of the overall text, it is awkward for this paragraph to suddenly appear.

-line 72~ Has this situation been explained to the patient and consent has been obtained?

-line 72~ In the process of selecting residents, how much experience do those residents have as orthopedic surgeons? If a resident who has little surgery experience and a chief resident are inevitably different. Shouldn't the level of experience be unified? This is a gross error in method.

-What do lines 88 to 89 mean?

-For line 105 ~ 108, present all the grades or present them in a table.

-Why didn't you check the versions of cup and stem in radiologic outcomes? Have you not evaluated CT, etc.? Radiologic outcomes are lacking.

-line 117 125, what level of significance did you set to be significant?

-For statistics, please describe which program you used.

Results

-It needs to be rewritten due to the need to modify the method.

Conlcusion

-Does the description on line 284 mean that this study design is wrong?

Author Response

Dear Reviewer,

Thank you very much for your comments and thoughts on this topic.

Best wishes,

the corresponding author.

Reviewer 2 Report

The authors have assessed the learning curve between the residents carrying out short stem total hip arthroplasty. They have observed significant differences between the residents.

1. It would provide additional value to the readers if additional factors that contribute to the learning curve are provided and compared between the residents such as their scores in courses during these years, their selection ranks during the first year of residency, their time in other surgical procedures.

2. Given the nature of variables and the numbers of cases, I think it is possible to use inferential statistics for comparing the difference between the residents. 

Author Response

(The authors gave the same response as above.)

Round 2

Reviewer 1 Report

Since the title mentions an anterolateral approach, please include a brief description of the surgical method.

Can this study (surgery results) be evaluated with only the radiologic outcome that measures only size, stem position - varus/valgus? How did you measure the version of the cup and stem?

Author Response

Dear Reviewer,

Thank you again for your quick response and comments.

I hope that the revision in the manuscript and my comment could again clarify your questions.

Best wishes,

The corresponding author.

Materials and Methods

Since the title mentions an anterolateral approach, please include a brief description of the surgical method.

We thank the reviewer for this comment and added a description in the manuscript.

Can this study (surgery results) be evaluated with only the radiologic outcome that measures only size, stem position - varus/valgus? How did you measure the version of the cup and stem?

Thank you for pointing this out. Not only radiological results have been evaluated as surgical results. Also surgical time, blood loos, length of stay, complications etc. were included, to have a variety of factors which could be representative for possible learning curve.

As it is not required from endoCERT Ltd. to measure the version of the cup and stem, this data was not available in our retrospective analysis. In general if we see highly pathologic versions of cups in postoperative x-rays and this results in clinical problems (especially joint dislocations), it will be recorded in the patient report. In our study population there were no reports like this and in particular no registered joint dislocations.

The measurement of the inclination of the cup was and is always done by the radiology department. An example is given below.

Best wishes, 

The corresponding author.
